# Effect of Ethylene *Sletr1-2* Receptor Allele on Flowering, Fruit Phenotype, Yield, and Shelf-Life of Four F1 Generations of Tropical Tomatoes (*Solanum lycopersicum* L.)

**Anas** [1,*], **Gungun Wiguna** [2], **Farida Damayanti** [1], **Syariful Mubarok** [1], **Dwi Setyorini** [2,*] and **Hiroshi Ezura** [3,4]

1   Department of Crop Science, Faculty of Agriculture, Universitas Padjadjaran, Bandung 45363, Indonesia
2   National Research and Innovation Agency the Republic of Indonesia, Jakarta 10340, Indonesia
3   Tsukuba Plant Innovation Research Center, University of Tsukuba, Tsukuba 305-8572, Japan
4   Faculty of Life and Environmental Sciences, University of Tsukuba, Tsukuba 305-8572, Japan
*   Correspondence: anas@unpad.ac.id (A.); dwis015@brin.go.id or rinibptpjatim@gmail.com (D.S.); Tel.: +62-812-8347-8305 (D.S.)

**Abstract:** A longer shelf-life for tomatoes without pleiotropic effects is one of the main goals of breeding programs in tropical countries. Therefore, this study aimed to evaluate the effect of the *Sletr1-2* mutant allele on flowering, fruit phenotype, shelf life, and yield-related traits in four F1 hybrids from four tropical tomato genetic backgrounds. The study consisted of four tropical strains, namely 'Intan', 'Mirah', 'Ratna', and 'Mutiara', as females crossed with wild type 'Micro-Tom' (WT-MT) and mutant *Sletr1-2* as males. Each was given three treatments and analyzed separately using a randomized block design with four replications of five samples each. The next test used was the Tukey Alpha 0.05 test. The genetic background of tropical tomatoes affects the phenotype and shelf-life. F1 mutants 'Intan' and 'Ratna' showed significant results, with a longer shelf-life than F1 WT (10.2 and 14.6 days, respectively). In addition, there were no side effects of the *Sletr1-2* mutant allele in the heterozygous form on flowering, fruit phenotype, and yield. In conclusion, the *Sletr1-2* allele has the potential to be used in tomato breeding programs in tropical countries.

**Keywords:** ethylene; shelf-life; *Sletr1-2*; tropical tomato; yield



## 1. Introduction

Tomato (*Solanum lycopersicum* L.) is one of the most popular vegetable crops in the world and is widely consumed by the public for health reasons. This fruit contains essential antioxidants such as lycopene, ascorbic acid, and phenolic compounds that help prevent chronic and coronary heart disease [1–3]. However, the short shelf-life of tomatoes is one of the main issues related to post-harvest quality, especially in tropical countries.

Tomatoes cultivated in Indonesia, a tropical island nation, take a long time to distribute to other regions. Therefore, a long shelf-life is the most demanded property in almost all horticultural products. The high yield loss during the post-harvest process has an economic impact on the incomes of farmers and traders. Increased shelf-life will result in lower post-harvest high-yield losses because fruit quality will be preserved, and damage will be reduced. One of the main factors affecting tomato fruit ripening is ethylene production [4].

Ethylene is a hormone important for plant growth and development [5–7]. This hormone plays a central role in several physiological and metabolic processes affecting the flavor and softness of the fruit [8,9]. Meanwhile, it has been reported that environmental factors such as temperature affect ethylene biosynthesis; high temperature increases ethylene production, which accelerates fruit softening [10,11]. Ethylene regulates many aspects of the plant life cycle, including seed germination, root formation, flower development, fruit ripening, senescence, and response to biotic and abiotic stresses [12]. As an aging

hormone, it promotes flowering, fruit ripening, leaf and petal senescence, pruning, and the plant's response to environmental stress [11,13,14]. Optimizing the plant response to ethylene promotes plant growth. It is now clear that ethylene plays a major role in the aging process of plants. A longer shelf-life for many fruits can be achieved by removing ethylene from the atmosphere surrounding the fruit [15]. Many biological processes take place after harvest, resulting in significant fluctuations between ethylene production and respiration in fruit and vegetables [16].

Tomato mutants with mutations in the ethylene receptor gene (*SlETR1*) were screened using the Micro-Tom TILLING platform [17–19]. The denatured high-performance liquid chromatography (*dHPLC*) method was first described as a native TILLING method that elicited *Arabidopsis* EMS mutant populations, DNA isolation and collection, PCR amplification of the target region, and heteroduplex formation and heteroduplex search [17]. Two mutant alleles, namely *Sletr1-1* and *Sletr1-2*, have been identified to reduce ethylene sensitivity and show dominant inheritance [17]. The first study to incorporate a mutated ethylene receptor gene (*SlETR1*) into subtropical tomatoes showed good fruit quality traits such as pH, fruit wall thickness, fruit firmness, and shelf-life [20]. The *Sletr1-2* mutant exhibited a moderate ethylene insensitivity phenotype, making the allele more suitable for breeding materials to extend fruit shelf-life [17,20,21]. Tests on subtropical tomatoes have been reported with the above results. Therefore, it is required to examine the impact of using this mutant on tropical tomatoes with various genetic backgrounds to determine their impact on fruit shelf life, flowering, and yield traits.

## 2. Materials and Methods

### 2.1. Development of F1 Populations

The parental tropical tomato varieties 'Intan', 'Mirah', 'Ratna', and 'Mutiara' were crossed with 'Micro-Tom' wild types (WT-MT) and 'Micro-Tom' mutants (*Sletr1-2*) to produce the F1 generation.

Meanwhile, the cultivars used as the parents for crossing are varied in several characters. 'Intan' and 'Ratna' have several superior phenotypes to 'Mirah' and 'Mutiara', i.e., more significant fruit weight and size and lower weight loss during 20 storage days (Table 1). Furthermore, the *Sletr1-2* mutant and the wild type 'Micro-Tom' (WT-MT) were used as male parents. In addition, four F1 *Sletr1-2* mutants of generations (F1 mutant), four F1 WT-MT generations (F1 WT), and their tropical tomato cultivars were evaluated.

**Table 1.** Description of four tropical tomato cultivars (female parents/♀).

| Variety | Origin | Growth Type | First Harvest (dap) | Fruit Weight (g) | Fruit Diameter (cm) | Pericarp Thickness (cm) | Fruit Weight Loss (20 dah) |
|---|---|---|---|---|---|---|---|
| 'Intan' | AVRDC Taiwan, China, introduction variety | Determinate | 80–85 | 40–65 | 3.8–5.0 | 0.3–0.4 | 4.15 |
| 'Mirah' | Indonesia local variety | Determinate | 75–85 | 20–40 | 3.0–4.0 | 0.2–0.3 | 6.68 |
| 'Ratna' | Philippine, introduction variety | Determinate | 80–85 | 45–85 | 5.0–6.0 | 0.3–0.4 | 5.17 |
| 'Mutiara' | Progeny of South American tomato | Indeterminate | 80–90 | 35–50 | 3.8–4.5 | 0.3–0.5 | 7.92 |

Note: dap = days after planting, dah = days after harvesting. AVRDC = Asian Vegetable Research and Development Center.

Data were recorded from twenty plants of each genotype, and the experiment was carried out in the screen-house at the vegetable Research Institute, Lembang, Indonesia, which is located at 1250 m above sea level (asl). Furthermore, temperature and humidity data were collected daily using Elitech RC-4&5 Conventional (V2.0) Temperature Data Logger Review (Elitech Ltd., London, UK).

Each plant was grown in a 40 cm diameter polybag containing soil and manure (1:1/*v:v*) mixture. Nitrogen–Phosphorus–Potassium compound fertilizer (15:15:15) was added at a dose of 2 g per plant. Furthermore, an additional 5 g Nitrogen–Phosphorus–Potassium per plant was added at 14, 30, 45, 60, 75, and 90 days after planting (DAP).

### 2.2. Confirmation of Mutant Hybrid

Cleaved Amplified Polymorphic Sequence (CAPS) markers were carried out according to [22] to determine the presence of the *Sletr1-2* mutant allele. Furthermore, DNA extraction was carried out using a method developed by [23]. DNA amplification by PCR was also conducted with a primer pair *SlETR1-CAPS* (forward): 5′-gtataaaaggagttggggcaaag-3′ and *SlETR1-CAPS* (reverse) 5′-atcaggaatgatgtggacaagc-3′. The total PCR reaction volume was 25 μL containing 5 μL of genomic DNA, Go Taq Green 12.5 μL forwards and reverse primers each with a concentration of 1 μL and 5.5 mL of Milli-Q water.

DNA amplification consisted of the following program: initial denaturation at 95 °C for 2 min, followed by 30 cycles of denaturation at 95 °C for 30 s, primer attachment at 57 °C for 30 s, and elongation/synthesis of DNA at 72 °C for 40 s. In the final extension, the PCR process was carried out at 72 °C for 6 min, and the PCR product was kept at 4 °C until analysis.

Each product was digested with the *MboI* enzyme (New England Biolabs, Inc., Ipswich, MA, USA) and incubated at 37 °C for 4 h. After digestion, each restriction-digested PCR product was subjected to electrophoresis and visualized in 2% agarose gels. Meanwhile, only plants that were identified as carrying the *Sletr1-2* allele were further evaluated. In addition, the molecular weights were analyzed by *PyElph* 1.4 software, as described in [24]

### 2.3. Evaluation of Fruit Shelf-Life

The fruit was harvested at the breaker stage + for 6 days (Br + 6), which was determined as 0 days of post-storage (DPS) for fruit shelf-life (FSL) analysis. Also, Br + 6 was characterized by >60% and <90% of changing red color on the fruit surface or at the light red stage [21]. Twenty fruits of each F1 generation were subjected to FSL evaluation in the laboratory at a mean 22 °C room temperature and a mean 77% humidity. Furthermore, FSL was determined by counting the number of days between the first day of storage (0 DPS) and a wrinkled appearance on the fruit surface. In addition, the extent of the relative fruit shelf-life (RFSL) of the F1 generation compared to their parent was determined using the formula: RFSL = (average of F1 generation FSL)/(average of the parent FSL) × 100%.

### 2.4. Evaluation of Flowering and Fruit Phenotypes

Observation of days to flowering (days) was recorded when 50% of the population blossomed. Furthermore, days to maturity (days) were determined by the first day of harvesting. Five fruits from each plant, which were harvested at Br + 6, were evaluated to elucidate the phenotypes. The fruit firmness was analyzed using a penetrometer (Precision Scientific Inc., Chicago, IL, USA). The diameter (cm) was measured at the fruit center using a vernier caliper. After dissecting each fruit's equatorial plane, the pericarp thickness (mm) was measured at two points using a vernier caliper. In addition, the locule numbers were recorded after the fruit was transversely cut.

### 2.5. Evaluation of Yield

The number of fruits and weight per plant were obtained from the harvested total. Also, six inflorescences per plant were maintained to produce fruit for the evaluation. Furthermore, the average weight per fruit was calculated from the weight per plant divided by the number of fruits per plant.

### 2.6. Statistical Analysis

The randomized block design used for this experiment consisted of four trials, namely four tropical tomato strains, and each experiment had three treatments, each with four replications and five treatment samples. There was a total of 12 treatments. A completely random block design was used to conduct a variance ANOVA analysis on observational data. The Tukey alpha 0.05 additional test was performed once the findings indicated a difference.

## 3. Results

### 3.1. Hybridization

Hybridization of the *Sletr1-2* mutant and tropical tomato cultivars proved effective. Hybridization was performed in March 2017 to produce F1 seeds for inspection, and each fruit had an average of 112 F1 seeds. The F1 generation was then assessed in August 2017, and the minimum/maximum temperature and humidity values of the assessment were 20.6 °C and 33.3 °C and 51.6% and 79.7%, respectively.

### 3.2. Transfer of the Sletr1-2 Allele

Molecular identification with CAPs markers showed that all F1 mutants had the mutant allele *Sletr1-2* (Figure 1). In addition, *MboI* is digested by F1 mutant plants into four fragments, namely 749 bp, 552 bp, 361 bp, and 152 bp. Furthermore, the molecular weight of tropical tomato cultivars showed fragments similar to 'Micro-Tom' (wild type) (Figure 1). These results demonstrate that hybridization between tropical tomato cultivars and the *Sletr1-2* mutant tomato was successfully conferred. In addition, the assessment of the F1 generation was performed based on a comparison between F1 and F1 WT mutants from the same female parent.

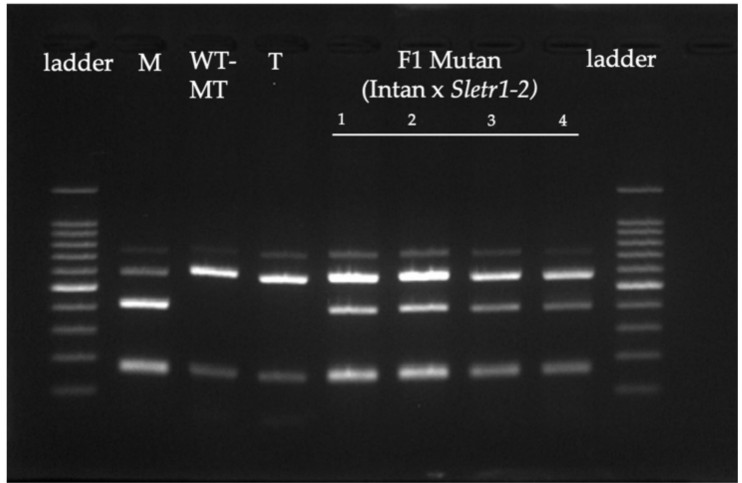

**Figure 1.** Genotyping of the *Sletr1-2* mutant allele in F1 mutant plants by CAPs marker. Ladder (DNA ladder 100 bp); M (mutant *Sletr1-2*); WT-MT (wild type 'Micro-Tom'); T (tropical tomato cultivar 'Intan'); F1 mutant (F1 plants from 'Intan' × *Sletr1-2* mutant). Genotyping of the *Sletr1-2* other tropical tomato cultivars ('Mirah', 'Ratna', and 'Mutiara') produces the same picture appearance.

### 3.3. Heterozygous Sletr1-2 Alleles

F1 'Intan' and 'Ratna' mutants showed significantly longer fruit shelf-life (FSL) compared to F1 WT, 10.2 days and 14.6 days, respectively (Figure 2), while F1 'Mirah' and 'Mutiara' showed no difference compared to F1 WT. However, all F1 mutant plants (F1 'Intan', F1 'Mirah', F1 'Ratna', and F1 'Mutiara') showed differences compared to the parent or parent strains ('Intan', 'Mirah', 'Ratna', and 'Mutiara'). The percentage difference in FSL of F1 'Intan' with its parent 'Intan' is 207.0%, and in F1 'Ratna' and its parent 'Ratna' is 265.2%. As shown in Figure 3, the appearance of wrinkling in fruit can occur as early as 20 DPS in all tropical tomato cultivars, while most F1 mutants exhibit such a fruit phenotype at 30 DPS.

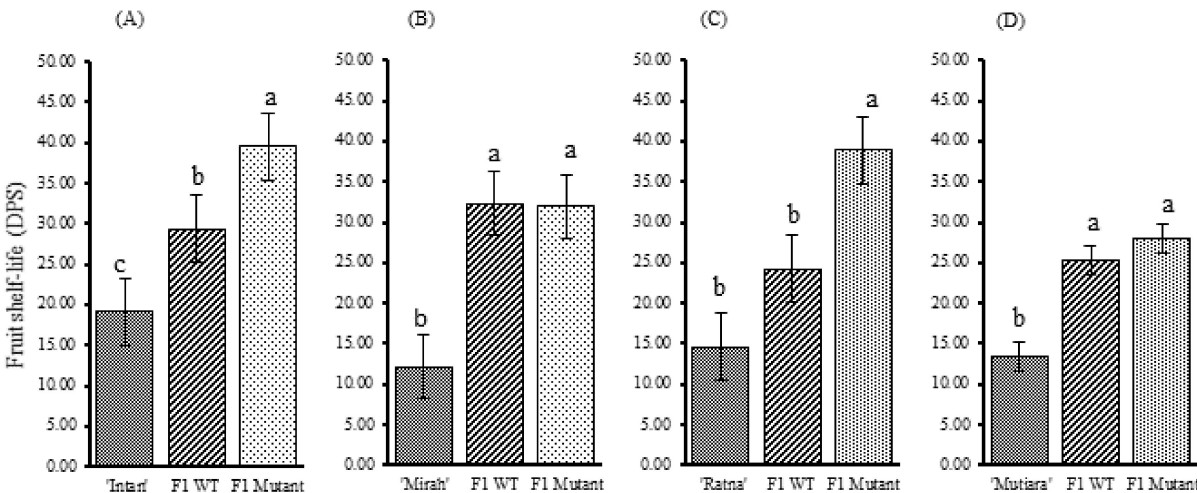

**Figure 2.** Shelf-life of F1 generation from four tropical tomato genetic backgrounds. (**A**) 'Intan', 'Intan' × 'WT', and 'Intan' × *Sletr1-2*; (**B**) 'Mirah', 'Mirah' × 'WT', and 'Mirah' × *Sletr1-2*; (**C**) 'Ratna', 'Ratna' × 'WT', and 'Ratna' × *Sletr1-2*; (**D**) 'Mutiara', 'Mutiara' × 'WT', and 'Mutiara' × *Sletr1-2*; Value is means ± SE (*n* = 4 × 5). Letters denote statistically distinguishable values as assessed by the HSD Tukey test *p* < 0.05.

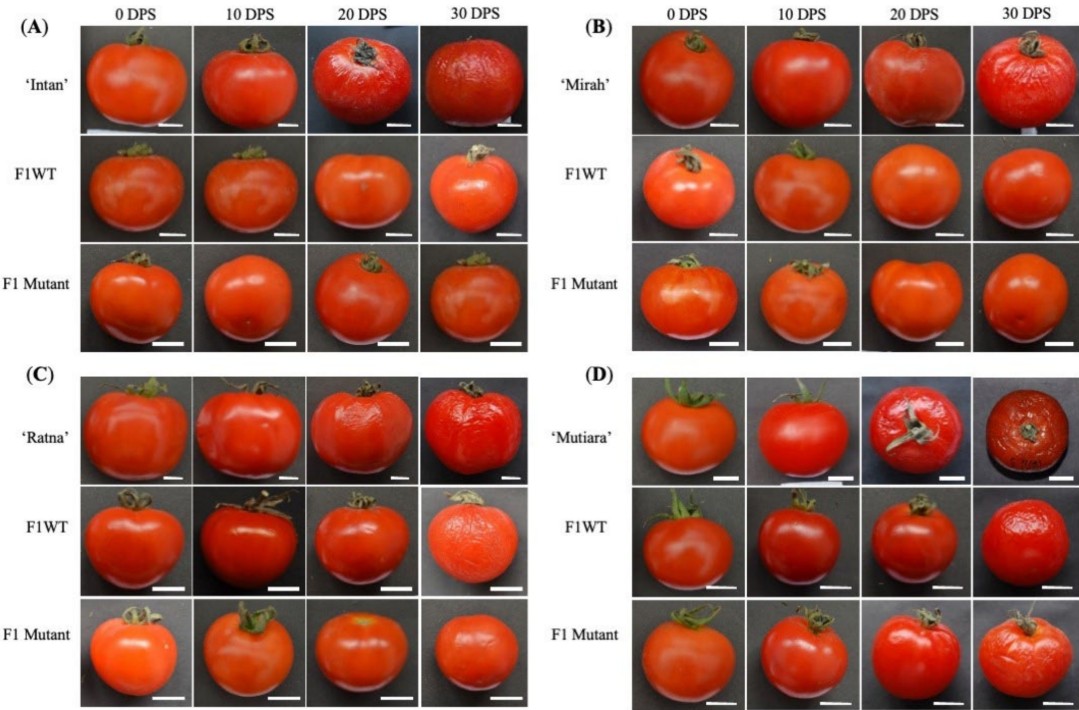

**Figure 3.** Fruit phenotypes of (**A**) 'Intan', 'Intan' × 'WT', and 'Intan' × *Sletr1-2*; (**B**) 'Mirah', 'Mirah' × 'WT', and 'Mirah' × *Sletr1-2*; (**C**) 'Ratna', 'Ratna' × 'WT', and 'Ratna' × *Sletr1-2*; (**D**) 'Mutiara', 'Mutiara' × 'WT', and 'Mutiara' × *Sletr1-2*; during the fruit shelf-life evaluation. The fruits were stored for 30 days under room temperature conditions at ±22 °C and ±77% humidity. Scale bars = 1 cm.

### 3.4. Effect of the Sletr1-2 Allele on Flowering and Fruit Phenotypes

In general, the *Sletr1-2* mutant allele did not affect fruit firmness, as mutants F1 and F1 WT did not show significant differences for this trait, except for the 'Mutiara' genetic background (Figure 4). This shows that fruit firmness is also influenced by the genetic background of tropical parents. In addition, the F1 'Mutiara' mutants showed significantly better fruit firmness than their F1 WT. (Figure 4).

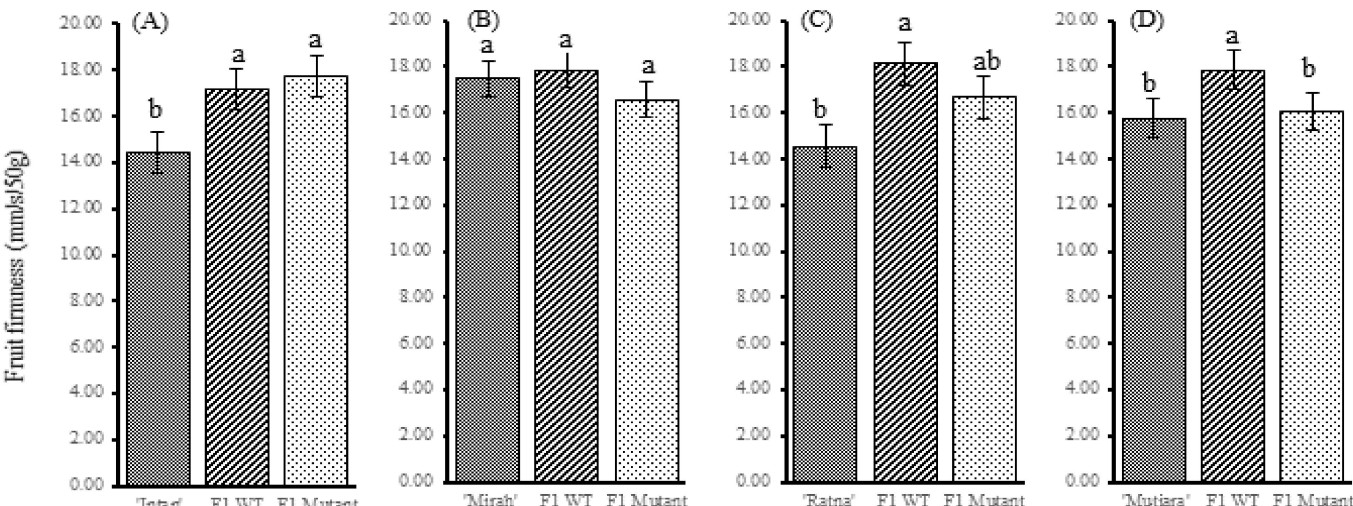

**Figure 4.** Fruit firmness of F1 generation from four tropical tomato genetic backgrounds. (**A**) 'Intan', 'Intan' × 'WT', and 'Intan' × *Sletr1-2*; (**B**) 'Mirah', 'Mirah' × 'WT', and 'Mirah' × *Sletr1-2*; (**C**) 'Ratna', 'Ratna' × 'WT', and 'Ratna' × *Sletr1-2*; (**D**) 'Mutiara', 'Mutiara' × 'WT', and 'Mutiara' × *Sletr1-2*; Value is means ± SE (*n* = 4 × 5). Letters denote statistically distinguishable values as assessed by the HSD Tukey test *p* < 0.05.

On the day of flowering, no difference was observed between the F1 WT and F1 mutants (Table 2). Similar results were also observed in days to maturity, with the exception of the 'Mutiara' genetic background. Meanwhile, both the F1 mutant and F1 WT showed significant flowering and early ripening compared to the respective tropical tomato cultivars (Table 2).

**Table 2.** Days to flowering and days to maturity of F1 generation in four tropical tomato genetic backgrounds.

| Genotype | Days to Flowering | Days to Maturity |
|---|---|---|
| 'Intan' (parent ♀) | 30.25 ± 0.75 [a] | 81.70 ± 0.68 [a] |
| 'Intan' F1 WT | 28.00 ± 0.41 [b] | 67.10 ± 0.77 [b] |
| 'Intan' F1 mutant | 28.00 ± 0.41 [b] | 65.60 ± 0.38 [b] |
| 'Mirah' (parent ♀) | 30.00 ± 0.71 [a] | 78.80 ± 2.81 [a] |
| 'Mirah' F1 WT | 28.50 ± 0.50 [ab] | 72.30 ± 1.52 [ab] |
| 'Mirah' F1 mutant | 28.00 ± 0.41 [b] | 67.45 ± 1.34 [b] |
| 'Ratna' (parent ♀) | 31.00 ± 0.41 [a] | 82.45 a ± 1.07 [a] |
| 'Ratna' F1 WT | 28.00 ± 0.41 [a] | 69.35 b ± 0.64 [b] |
| 'Ratna' F1 mutant | 28.00 ± 1.22 [a] | 68.70 b ± 1.88 [b] |
| 'Mutiara' (parent ♀) | 34.75 ± 1.70 [a] | 84.65 a ± 1.45 [a] |
| 'Mutiara' F1 WT | 29.00 ± 0.58 [b] | 75.65 b ± 1.58 [b] |
| 'Mutiara' F1 mutant | 28.00 ± 0.41 [b] | 69.00 c ± 1.39 [c] |

Values are means ± SE (*n* = 4 × 5). Values followed by the same letters are not significantly different according to Tukey's HSD test at *p* < 0.05 in each genetic background.

The results indicated that the presence of the *Sletr1-2* allele, in general, had no pleiotropic effect on the fruit phenotype observed in this study. This was because there was no significant difference between the F1 WT and F1 mutants in the genetic background (Table 3). The observed differences between F1 WT and F1 mutant fruit phenotypes appeared to be more influenced by the genetic background of tropical tomato cultivars.

**Table 3.** Fruit phenotypes of F1 generation in four tropical tomato genetic backgrounds.

| Genotype | Fruit Diameter (cm) | Locule Number | Pericarp Thickness (mm) |
|---|---|---|---|
| 'Intan' (parent ♀) | 4.37 ± 0.14 [a] | 4.60 ± 0.18 [a] | 3.35 ± 0.16 [a] |
| 'Intan' F1 WT | 4.16 ± 0.11 [a] | 3.55 ± 0.16 [b] | 3.04 ± 0.12 [ab] |
| 'Intan' F1 mutant | 3.48 ± 0.13 [b] | 4.25 ± 0.27 [ab] | 2.79 ± 0.12 [b] |
| 'Mirah' (parent ♀) | 3.78 ± 0.15 [a] | 4.33 ± 0.21 [a] | 2.65 ± 0.13 [a] |
| 'Mirah' F1 WT | 3.71 ± 0.14 [a] | 4.00 ± 0.17 [a] | 2.80 ± 0.11 [a] |
| 'Mirah' F1 mutant | 3.58 ± 0.11 [a] | 4.07 ± 0.12 [a] | 2.91 ± 0.09 [a] |
| 'Ratna' (parent ♀) | 5.54 ± 0.19 [a] | 5.93 ± 0.27 [a] | 3.70 ± 0.21 [a] |
| 'Ratna' F1 WT | 3.53 ± 0.08 [b] | 4.09 ± 0.14 [b] | 3.01 ± 0.14 [b] |
| 'Ratna' F1 mutant | 3.30 ± 0.07 [b] | 3.85 ± 0.95 [b] | 2.65 ± 0.10 [b] |
| 'Mutiara' (parent ♀) | 4.25 ± 0.15 [a] | 3.54 ± 0.20 [a] | 3.98 ± 0.17 [a] |
| 'Mutiara' F1 WT | 3.28 ± 0.08 [b] | 3.54 ± 0.15 [a] | 2.70 ± 0.11 [b] |
| 'Mutiara' F1 mutant | 3.57 ± 0.11 [b] | 4.08 ± 0.23 [a] | 2.64 ± 0.13 [b] |

Values are means ± SE ($n = 4 \times 5$). Values followed by the same letters are not significantly different according to Tukey's HSD test at $p < 0.05$ in each genetic background.

### 3.5. Effect of the Sletr1-2 Allele on Yield-Related Traits

In general, the number of fruits and weight per plant of the F1 mutant was not significantly different from that of the two F1 WT, except for the genetic background of 'Mutiara' (Table 4). It was hypothesized that no pleiotropic effect of the *Sletr1-2* allele on outcome-related traits was observed in the F1 mutant. In addition, the F1 WT and F1 mutants showed significantly lower fruit weights than the corresponding tropical tomato cultivars. However, the lower fruit weight was not the only factor determining the final yield, as shown by the genetic background of 'Mirah' and 'Ratna'.

**Table 4.** Yield-related traits of $F_1$ generation in four tropical tomato genetic backgrounds.

| Genotype | Fruit Weight (g) | Number of Fruit per Plant | Fruit Weight per Plant (g) |
|---|---|---|---|
| 'Intan' (parent ♀) | 53.28 ± 3.07 [a] | 27.30 ± 1.41 [b] | 1287.52 ± 82.52 [a] |
| 'Intan' F1 WT | 18.24 ± 0.88 [b] | 47.30 ± 3.45 [a] | 847.05 ± 69.84 [b] |
| 'Intan' F1 mutant | 18.08 ± 0.64 [b] | 45.20 ± 2.94 [a] | 719.45 ± 51.05 [b] |
| 'Mirah' (parent ♀) | 30.21 ± 4.06 [a] | 44.40 ± 4.38 [a] | 1105.45 ± 84.53 [a] |
| 'Mirah' F1 WT | 21.63 ± 2.29 [b] | 47.90 ± 3.62 [a] | 959.90 ± 76.76 [a] |
| 'Mirah' F1 mutant | 23.58 ± 3.05 [b] | 44.85 ± 3.63 [a] | 933.45 ± 61.24 [a] |
| 'Ratna' (parent ♀) | 67.10 ± 6.20 [a] | 27.05 ± 1.91 [b] | 1391.20 ± 102.55 [a] |
| 'Ratna' F1 WT | 17.95 ± 1.15 [b] | 49.95 ± 2.96 [a] | 823.90 ± 56.44 [a] |
| 'Ratna' F1 mutant | 15.73 ± 0.86 [b] | 50.60 ± 3.13 [a] | 723.35 ± 69.06 [a] |
| 'Mutiara' (parent ♀) | 42.97 ± 2.62 [a] | 36.75 ± 3.01 [c] | 1394.65 ± 96.74 [a] |
| 'Mutiara' F1 WT | 19.23 ± 0.95 [b] | 54.20 ± 4.70 [b] | 940.35 ± 84.13 [b] |
| 'Mutiara' F1 mutant | 21.00 ± 1.46 [b] | 69.35 ± 4.32 [a] | 1214.55 ± 117.26 [ab] |

Values are means ± SE ($n = 4 \times 5$). Values followed by the same letters are not significantly different according to Tukey's HSD test at $p < 0.05$ in each genetic background.

### 4. Discussion

The mutated allele *Sletr1-2* was successfully crossed into tropical tomato varieties by conventional crossing (Figure 1). This study confirmed that the *Sletr1-2* allele extended the shelf-life of F1 plants. However, the effect of the *Sletr1-2* allele on fruit shelf-life depends on the genetic background of tropical tomatoes (Figure 2). This indicates that the cultivar crossed with the *Sletr1-2* allele plays an important role in the expression of this gene and determines the extent of the resulting shelf-life extension. In [20], the authors also found that diversity and the presence of ripening-inhibiting genes contribute to fruit shelf-life

and post-harvest quality dynamics, and [21] reported that the genetic background of the parents influenced the extension of the shelf-life of fruits in the gene disrupting method using the nor gene. The same phenomenon occurred in a study using the *alc* gene to extend the shelf-life of tomatoes [24]. Judging by flowering, fruit phenotype, and yield, with the exception of F1 'Mutiara', which matured later than F1 WT and parent plant 'Mutiara', which was an indeterminate tomato variety (Table 1).

The tropical parents used in this study differed in several characteristics (Table 1). 'Intan' and 'Ratna' compared to 'Mirah' and 'Mutiara' are larger, have more weight, and have less weight loss during storage. A lower fruit weight loss also indicates that the variety has a slow ripening process. This variability is thought to influence the increase in the shelf-life of fruit by the *Sletr1-2* allele. Therefore, the inheritance of this character affects the extension of the shelf-life of the fruit from the offspring.

In this study, F1 mutants from all cross combinations showed a longer shelf-life than any tropical tomato variety. In contrast, ref. [18] reported shorter shelf-life performance in the F1 mutant compared to its subtropical parent. It has also been suggested that different gene actions might occur between tropical and subtropical tomatoes on the mutated allele *Sletr1-2*. On the other hand, the extended shelf-life of the F1 mutant may be due to the interaction of advantageous alleles at the heterozygous locus [25].

Extending shelf-life can reduce high-yield losses in the tomato supply chain in Indonesia. The vegetable supply chain in Indonesia is quite long, involving five to six economic actors dominated by traditional supply chains [26]. During the distribution process, the damage to agricultural raw materials is relatively large, and almost 70% of plant products become waste. In addition, tomatoes are the second largest horticultural product to be perishable (perishable) after potatoes. Without proper treatment, tomatoes lose their firmness within a few days, look wrinkled, and then rot. In addition, the amount of wasted tomatoes reduces farmers' profits. In addition, distribution costs become more wasteful as the products shipped are damaged and eventually discarded. These high losses inevitably lead to an increase in logistics costs.

Several factors are closely related to fruit elasticity, including water content and ripeness. Fruits with high water content tend to be softer than fruits with low water content [27]. Likewise, faster-ripening fruit yields softer fruit because ethylene accelerates the hydrolysis of polysaccharides in the cell wall and increases water content during the ripening process [28,29]. The F1 'Intan' and 'Ratna' mutants showed a lower fruit hardness measured at Br+6 compared to the F1 'Mirah' and 'Mutiara' mutants. However, the first two genetic backgrounds showed longer shelf-life, suggesting that the presence of the *Sletr1-2* allele in both F1 mutants had slowed the fruit ripening process and prevented an increase in water content. Such a mechanism could help extend the shelf-life of fruit [17].

In addition, there is a genetic contribution of tropical tomatoes. 'Intan' has a longer shelf-life of 22.25 days after harvest and 'Ratna' has a shelf-life of 21.5 days after harvest [30], which is longer than 'Mirah' with a shelf-life of 9 days, and 'Mutiara' [31,32]. The *Sletr1-2* mutant allele did not affect the days to flowering in the F1 generations. Moreover, there was no difference in days to flowering between F1 mutant and F1 WT in all tropical tomato genetic backgrounds (Table 1). Ref. [22] reported that the *Sletr1-2* mutant allele has weak ethylene insensitivity; hence, this allele does not entirely block the ethylene effect. Furthermore, Ref. [5] stated that ethylene impacts in flowering are complex. The presence of ethylene in some types of plants shows an inhibiting process of flowering, such as in *Arabidopsis* [33], while a contrasting response is shown by other plants, such as in pineapple [5].

Fruit phenotypes, namely fruit wall thickness and the number of loci, were unaffected by the presence of the *Sletr1-2* allele in the F1 mutant in this study (Table 3). The number of compartments and the thickness of the pericarp were also strongly dependent on the size of the fruit [34,35]. This variation is genotypic, and the maturation mutant does not affect this trait [25].

The *Sletr1-2* allele in F1 mutant plants did not affect the yield-related traits observed in this study. As in other maturing mutant genes such as *Alcobaça (alc)*, the gene in the

heterozygous form had no effect on total yield or mean fruit mass [36]. Additionally, lower fruit weights in both F1 mutants and F1 WT compared to the respective tropical tomato varieties were mainly due to the genetic background of 'Micro-Tom' showing small plants and small fruit size [18].

## 5. Conclusions

No pleiotropic effect of the *Sletr1-2* allele was observed on flowering, fruit phenotype, and yield, except in F1 'Mutiara', which ripened later than F1 WT, and the parent 'Mutiara', which was an indeterminate type tomato. It was concluded that the *Sletr1-2* mutant could be utilized in the development of new cultivars with better shelf-life in tropical countries by considering the genetic background of the tropical tomatoes used as parents; in this study, the parents 'Intan' and 'Ratna'. Further testing is needed on the effect of the *Sletr1-2* mutant on the quality of tomatoes in more detail on the nutritional content of tomatoes.

**Author Contributions:** Conceptualization, A., G.W., S.M. and H.E.; Methodology, G.W. and A.; software, G.W.; validation, A., S.M. and F.D.; formal analysis, G.W. and A.; investigation, G.W., A. and S.M.; resources, G.W., A. and S.M.; data curation, G.W., A. and F.D.; writing-preparation of original draft, A., G.W., F.D. and D.S., writing—reviews and editing, A., F.D. and D.S.; visualization, G.W., F.D. and D.S.; supervision, A. and H.E.; project administration, A.; fundraising, A. All authors have read and agreed to the published version of the manuscript.

**Funding:** This research was funded by RKDU research grant Universitas Padjadjaran grant number 1959/UN6.3.1/PT.00/2021. The APC was funded by Universitas Padjadjaran.

**Institutional Review Board Statement:** The study was conducted in accordance with the Declaration of Helsinki, and approved by the Padjadjaran, University.

**Informed Consent Statement:** Not applicable.

**Data Availability Statement:** The study did not report any data.

**Acknowledgments:** To the Indonesian Agency Agricultural for Research and Development, Ministry of Agriculture, Republic of Indonesia, which has given the opportunity to carry out this research activity.

**Conflicts of Interest:** The authors declare no conflict of interest.

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
