# Peer review of "Effect of Ethylene Sletr1-2 Receptor Allele on Flowering, Fruit Phenotype, Yield, and Shelf-Life of Four F1 Generations of Tropical Tomatoes (Solanum lycopersicum L.)"

_horticulturae, doi:10.3390/horticulturae8121098_

Round 1

Reviewer 1 Report

The authors cross-bred tomatoes (Solanum lycopersicum) using four tomato cultivars (‘Intan’, ‘Mirah’, ‘Ratna’, or ‘Mutiara’) as the mother plant and Sletr1-2 mutant or wild type ‘Micro Tom’ as the father plant. Then, the authors evaluated and compared serval parameters regarding flowering period, fruit phenotype, yield, and shelf-life of F1 hybrids within a certain cultivar. They found that shelf-life was prolonged and no detrimental effects on flowering, fruit phenotype as well as yield in the F1 hybrids using Sletr1-2 mutant. The results suggest using Sletr1-2 mutant for tomato breeding program to prolongate its shelf-life.

The authors successfully developed prolonged shelf-life tropical tomatoes by hybridization, which was a great achievement. Hopefully, these four hybrids could be well commercialized in Indonesia or other tropical areas. However, before this research work got submitted to horticulturae, the effect of Sletr1-2 mutation/ethylene insensitivity in tomatoes on their fruit shelf life has been well studied*. The results presented in this submission may not be able to provide advanced knowledge. Almost all these publications focused on the same tomato species as in this submission, Solanum lycopersicum, therefore, besides the rational in L61-62, the authors of this manuscript might need to address the question ‘what is new (beyond tropical, subtropical, and temperate tomatoes)’.

Tomato fruits are climacteric, and they are sensitive to ethylene presence. This feature significantly impacts their shelf life. As a widely consumed commodity, contents of sugars, organic acids, amino acids, and carotenoid in newly developed tomato cultivars and their pericarp thickness which impact the flavor, nutrient value, and texture are essential parameters to determine whether they will be able to be accepted by consumers. Therefore, some further investigations for the hybrids in this manuscript were expected by the reviewer to promote this research an amazing story, including but not limited to determinations of ethylene production, content analyses of the compounds mentioned above.

*References: 

·       Mubarok, S., Okabe, Y., Fukuda, N., Ariizumi, T. and Ezura, H., 2015. Potential use of a weak ethylene receptor mutant, Sletr1-2, as breeding material to extend fruit shelf life of tomato. Journal of Agricultural and Food Chemistry63(36), pp.7995-8007. DOI: https://doi.org/10.1021/acs.jafc.5b02742

·       Mubarok, S., Okabe, Y., Fukuda, N., Ariizumi, T. and Ezura, H., 2016. Favorable effects of the weak ethylene receptor mutation Sletr1-2 on postharvest fruit quality changes in tomatoes. Postharvest biology and technology120, pp.1-9. DOI: https://doi.org/10.1016/j.postharvbio.2016.04.022

·       Mubarok, S., Ezura, H., Rostini, N., Suminar, E. and Wiguna, G., 2019. Impacts of Sletr1-1 and Sletr1-2 mutations on the hybrid seed quality of tomatoes. Journal of Integrative Agriculture18(5), pp.1170-1176. DOI: https://doi.org/10.1016/S2095-3119(19)62614-6

·       Mubarok, S., Hoshikawa, K., Okabe, Y., Yano, R., Tri, M.D., Ariizumi, T. and Ezura, H., 2019. Evidence of the functional role of the ethylene receptor genes SlETR4 and SlETR5 in ethylene signal transduction in tomato. Molecular Genetics and Genomics294(2), pp.301-313. DOI: https://doi.org/10.1007/s00438-018-1505-7

Figures and Tables, including captions, should be described in a much clearer manner to make it easy to follow or avoid misunderstanding. 

Please note that the examples were offered below as comprehensively as possible. However, they are not intended to represent an accurate, comprehensive list of the issues that need to be tackled. Please change the first occurrence of each problem and check the entire manuscript for similar or related ones.

INTRODUCTION

Major comments:

The authors might want to add more citations when mentioning “people extensively consume it for health purposes.”, “the short shelf-life of tomato fruits is one of the significant problems related to post-harvest quality, especially in tropical countries”.

The reviewer was not sure the meaning of the sentences “Tomato cultivation in high temperature and high humidity conditions … can significantly reduce the quality.” In L34-35 and “…high yield losses during post-harvest economically decrease farmers’ and traders’ incomes” in L42-43. The authors might want to rephrase this one to provide a better logic flow.

Again, as the authors mentioned, “One of the significant factors affecting fruit maturity in tomatoes is ethylene production in tomatoes” in L43-44, determining the ethylene production in their F1 hybrid fruits was expected by the reviewer, compared with evaluating their flowering period. 

Gene name should be italicized. Please re-confirm the whole manuscript to keep it consistently correct. For example, SlETR1 in L55&59, Sletr1-1 and Sletr1-2 in L57, 59, 64, 65, 78, 79, 107, 137, 149, 153, 157, 159, and 181.

Minor comments:

The whole name of the gene and its abbreviation was first provided in L55, the authors might want to simply use the abbreviation in L58-59. 

A period ‘.’ should be added right after the citations “(16-18)” in L61. 

M&M

Major comments:

The authors did not mention if they extracted DNA from all mother plants of the tomato cultivars but only presented the gel result about ‘Intan’ cultivar (labeled as ‘T’) and the hybrids between Sletr1-2 mutant x ‘Intan’ (labeled as ‘F1 Mutant’) in Fig. 1, which was confusing. Please clarify a little bit. 

Besides, to follow the general notation manner and avoid any misunderstanding, the mother plants, ‘Intan’, ‘Mirah’, ‘Ratna’, and ‘Mutiara’, should be placed before the notation ‘x’, while the father plants, ‘Micro-Tom’ and Sletr1-2mutant, should be placed after the notation ‘x’. The authors might want to double-check to keep the notation format consistent in the entire manuscript, including the figures and tables. For example, Sletr1-2 mutant x ‘Intan’ in the Fig. 1 caption V.S. ‘Intan’ x Sletr1-2 labelled in Fig. 3-A.

The reviewer wondered whether 1) the assumptions were checked before using ANOVA with Tukey’s HSD to analyze the data, and 2) the same parameter was assessed and compared among different F1 hybrid cultivars. 

Additionally, the experiment was RCB designed with four replications and ‘the data were represented as the mean values of four replicates’. The differences in each analysis parameter were separated among the mother plant, F1 hybrid of mother plant x wild type, and F1 hybrid of mother plant x Sletr1-2 mutant within the same cultivar in ‘Statistical analysis’ section. The n should be 12. However, it was shown as n=20 in L173, 188, 204, and 215. Please clarify the statistical analysis methods. 

Minor comments:

It should be ‘Determinate’, not ‘Determi-nate’ in the third row of Table 1 in L73.

In addition, what is ‘AVRDC’ short for?

‘Introduced variety’ in Table 1 seems more grammatically proper.

What is ‘the screen house’ in L82? 

The authors might want to provide the whole name for ‘NPK’ when mentioning the fertilizer in the first time in L87.

The authors might want to double-check the units for temperature in L102&103.

It should be ‘fruit shelf-life (FSL)’, not ‘shelf-life (FSL)’ in L112. Similarly, it should be ‘day of post-storage (DPS)’, not ‘day of storage (DPS)’ in L116.

 22’ and ‘ 77’ in L115&178 look so weird.

‘average FSL of F1’ and ‘average SFL of the parent’ seem more grammatically proper in L119. 

‘at’ should be added between ‘test’ and ‘p’ and ‘p’ should be italicized in L138. Please double-check the entire manuscript for the similar issue.

RESULTS

Major comments:

The authors did not mention if they extracted DNA from all mother plants of the tomato cultivars but only presented the gel result about ‘Intan’ cultivar (labeled as ‘T’) and the hybrids between Sletr1-2 mutant x ‘Intan’ (labeled as ‘F1 Mutant’) in Fig. 1, which was confusing. Please clarify a little bit. If the four ladders on the gel in Fig. 1 represented four different hybrids between the four cultivars and Sletr1-2 mutant, respectively. Please ensure the labels in the figure/table and descriptions in figure captions are shown in a clearer manner. 

Besides, figures were not notated by A, B, C, or D in Figs 2, 3, and 4. The authors might want to label them clearly in the figures. 

In addition, ‘F1 mutant’ and ‘F1 WT’ in Figs 1, 2, 3, and 4, as well as Tables 2, 3, and 4 were confusing. Highly recommend using F1 hybrid between [a certain cultivar] and Sletr1-2 mutant to enhance the readability. 

The Supplementary Table 1 or 2 was not found. The authors might want to double-check if these files were uploaded correctly.

Minor comments:

A period ‘.’ should be deleted after ‘Figure’ in L152.

‘at’ should be added between ‘test’ and ‘p’ and ‘p’ should be italicized in L174. Please double-check the entire manuscript for the similar issue.

There should be a space between the number and letter in the second column of Table 4.

REFERENCE

1) The authors may want to double-check the proper citation of the references. For example, reference #4. This cited paper did not actually research or conclude as “high yield losses during post-harvest economically decrease farmers’ and traders’ incomes”, as the authors mentioned here. In addition, https://www.sciencedirect.com/science/article/pii/S0223523403001004, the link shown in reference #4 was a review paper about inhibitors of Na+/H+ exchanger, instead of ethylene management to extend the shelf life of tomatoes.

2) This manuscript’s reference format deviates from Horticulturae format. Besides, the citations in the text should be placed in square brackets, instead of parenthesis. The authors may want to double-check the guidelines and modify the reference section accordingly. Here’s the guideline for the authors’ reference.

  • References: References must be numbered in order of appearance in the text (including table captions and figure legends) and listed individually at the end of the manuscript. We recommend preparing the references with a bibliography software package, such as EndNoteReferenceManager or Zotero to avoid typing mistakes and duplicated references. We encourage citations to data, computer code and other citable research material. If available online, you may use reference style 9. below. 
  • Citations and References in Supplementary files are permitted provided that they also appear in the main text and in the reference list.

In the text, reference numbers should be placed in square brackets [ ], and placed before the punctuation; for example [1], [1–3] or [1,3]. For embedded citations in the text with pagination, use both parentheses and brackets to indicate the reference number and page numbers; for example [5] (p. 10). or [6] (pp. 101–105).

The reference list should include the full title, as recommended by the ACS style guide. Style files for Endnote and Zotero are available.

References should be described as follows, depending on the type of work:

ï‚·  Journal Articles:
1. Author 1, A.B.; Author 2, C.D. Title of the article. Abbreviated Journal Name YearVolume, page range.

ï‚·  Books and Book Chapters:
2. Author 1, A.; Author 2, B. Book Title, 3rd ed.; Publisher: Publisher Location, Country, Year; pp. 154–196.
3. Author 1, A.; Author 2, B. Title of the chapter. In Book Title, 2nd ed.; Editor 1, A., Editor 2, B., Eds.; Publisher: Publisher Location, Country, Year; Volume 3, pp. 154–196.

ï‚·  Unpublished materials intended for publication:
4. Author 1, A.B.; Author 2, C. Title of Unpublished Work (optional). Correspondence Affiliation, City, State, Country. year, status(manuscript in preparationto be submitted).
5. Author 1, A.B.; Author 2, C. Title of Unpublished Work. Abbreviated Journal Name year, phrase indicating stage of publication(submittedacceptedin press).

ï‚·  Unpublished materials not intended for publication:
6. Author 1, A.B. (Affiliation, City, State, Country); Author 2, C. (Affiliation, City, State, Country). Phase describing the material, year. (phase: Personal communication; Private communication; Unpublished work; etc.)

ï‚·  Conference Proceedings:
7. Author 1, A.B.; Author 2, C.D.; Author 3, E.F. Title of Presentation. In Title of the Collected Work (if available), Proceedings of the Name of the Conference, Location of Conference, Country, Date of Conference; Editor 1, Editor 2, Eds. (if available); Publisher: City, Country, Year (if available); Abstract Number (optional), Pagination (optional).

ï‚·  Thesis:
8. Author 1, A.B. Title of Thesis. Level of Thesis, Degree-Granting University, Location of University, Date of Completion.

ï‚·  Websites:
9. Title of Site. Available online: URL (accessed on Day Month Year).
Unlike published works, websites may change over time or disappear, so we encourage you create an archive of the cited website using a service such as WebCite. Archived websites should be cited using the link provided as follows:
10. Title of Site. URL (archived on Day Month Year).

The authors might want to double-check the format for the ‘Author Contributions’, ‘Acknowledgements’ and ‘Conflicts of Interest’ sections to better follow the horticulturae guidelines.

Author Response

Dear.
Reviewer 1

We have corrected the manuscript according to your suggestions, hopefully, it will be what you want
Thank you

Author
Dwi Setyorini

Reviewer 2 Report

The manuscript is written with clear understanding of the project addressed. However, there are some concerns that need to be addressed to enhance the quality of the manuscript. My specific comments are as follows:

Abstract:

Elaborate more on methods of your study.

Introduction:

Add literatures on effect of ethylene on shelf life of fruit/vegetables

“Tomato mutants with mutations in the ethylene receptor gene (SlETR1) have been screened using a Micro-Tom TILLING platform.” Add citation

Based on your objectives, please compare how your study is different from those that have already been published

Materials and methods:

How many tomato samples used?

Elaborate more on the shelf life evaluation

Results and discussion:

Revise the title for all subsection in section 3 (Results)-should be in general form

Elaborate more on results for subsection 3.1

Figure 2 is not clear. Change to new image

“This is because there was no significant difference between F1 WT and F1 mutant in the genetic backgrounds (Table 3).” Elaborate on the no significant difference between mutant

The findings lack in terms of justification and major findings.

Conclusions:

Emphasize more on the main finding.

Add recommendation for future studies

General comments:

Please check the reference styles and grammar of the manuscript.

Author Response

Dear.
Reviewer 2

We have corrected the manuscript according to your suggestions, hopefully it will be what you want
Thank you

Author
Dwi Setyorini

Round 2

Reviewer 1 Report

The authors polished the manuscript significantly by rephrasing the sentences in Introduction section and captions in Figures & Tables, so these sections have a better logic flow. However, there are still some major comments that the authors did not respond specifically point by point. 

1. Before this research work got submitted to horticulturae, the effect of Sletr1-2 mutation/ethylene insensitivity in tomatoes on their fruit shelf life has been well studied*. The results presented in this submission may not be able to provide advanced knowledge. Almost all these publications focused on the same tomato species as in this submission, Solanum lycopersicum, therefore, besides the rational in L61-63, the authors of this manuscript might need to address the question ‘what is new (beyond tropical, subtropical, and temperate tomatoes)’.

*References: 

·       Wiguna, G., Damayanti, F., Mubarok, S., Ezura, H. and Anas, A.N.A.S., 2021. Genetic control of fruit shelf life and yield in crossbreeding of Sletr1-2 mutant with Indonesian tropical tomatoes: Combining Ability of Sletr1-2 mutant. Biodiversitas Journal of Biological Diversity, 22(10). DOI: https://doi.org/10.13057/biodiv/d221060

·       Mubarok, S., Okabe, Y., Fukuda, N., Ariizumi, T. and Ezura, H., 2015. Potential use of a weak ethylene receptor mutant, Sletr1-2, as breeding material to extend fruit shelf life of tomato. Journal of Agricultural and Food Chemistry63(36), pp.7995-8007. DOI: https://doi.org/10.1021/acs.jafc.5b02742

·       Mubarok, S., Okabe, Y., Fukuda, N., Ariizumi, T. and Ezura, H., 2016. Favorable effects of the weak ethylene receptor mutation Sletr1-2 on postharvest fruit quality changes in tomatoes. Postharvest biology and technology120, pp.1-9. DOI: https://doi.org/10.1016/j.postharvbio.2016.04.022

·       Mubarok, S., Ezura, H., Rostini, N., Suminar, E. and Wiguna, G., 2019. Impacts of Sletr1-1 and Sletr1-2 mutations on the hybrid seed quality of tomatoes. Journal of Integrative Agriculture18(5), pp.1170-1176. DOI: https://doi.org/10.1016/S2095-3119(19)62614-6

·       Mubarok, S., Hoshikawa, K., Okabe, Y., Yano, R., Tri, M.D., Ariizumi, T. and Ezura, H., 2019. Evidence of the functional role of the ethylene receptor genes SlETR4 and SlETR5 in ethylene signal transduction in tomato. Molecular Genetics and Genomics294(2), pp.301-313. DOI: https://doi.org/10.1007/s00438-018-1505-7

2. Tomato fruits are climacteric, and they are sensitive to ethylene presence. This feature significantly impacts their shelf life. As a widely consumed commodity, contents of sugars, organic acids, amino acids, and carotenoid in newly developed tomato cultivars and their pericarp thickness which impact the flavor, nutrient value, and texture are essential parameters to determine whether they will be able to be accepted by consumers. Therefore, some further investigations for the hybrids in this manuscript were expected by the reviewer to promote this research an amazing story, including but not limited to determinations of ethylene production, content analyses of the compounds mentioned above.

Please note that the examples were offered below as comprehensively as possible. However, they are not intended to represent an accurate, comprehensive list of the issues that need to be tackled. Please change the first occurrence of each problem and check the entire manuscript for similar or related ones.

INTRODUCTION

Major comments:

Again, as the authors mentioned, “One of the significant factors affecting fruit maturity in tomatoes is ethylene production in tomatoes” in L42-43, determining the ethylene production in their F1 hybrid fruits was expected by the reviewer, compared with evaluating their flowering period. 

Minor comment:

The period ‘.’ should be deleted in L68. 

M&M

Major comments:

The reviewer wondered whether 1) the assumptions were checked before using ANOVA with Tukey’s HSD to analyze the data, and 2) the same parameter was assessed and compared among different F1 hybrid cultivars. 

RESULTS

Major comments:

The Supplementary Table 1 or 2 was not found. The authors might want to double-check if these files were uploaded correctly.

Minor comments:

‘at’ should be added between ‘test’ and ‘p’ and ‘p’ should be italicized in L193. Please double-check the entire manuscript for the similar issue.

REFERENCE

1) It seems like that the authors did not double-check the proper citation of the references. For example, as the reviewer previously mentioned, the link for reference #4. https://www.sciencedirect.com/science/article/pii/S0223523403001004, the link was a review paper about inhibitors of Na+/H+ exchanger, instead of ethylene management to extend the shelf life of tomatoes. It was NOT the paper the authors intended to refer to.

2) This manuscript’s reference format still deviates from Horticulturae format. Please go through the guidelines attached below and modify the reference carefully using Endnote or Zotero.

The reference list should include the full title, as recommended by the ACS style guide. Style files for Endnote and Zotero are available.

References should be described as follows, depending on the type of work:

ï‚·  Journal Articles:
1. Author 1, A.B.; Author 2, C.D. Title of the article. Abbreviated Journal Name YearVolume, page range.

ï‚·  Books and Book Chapters:
2. Author 1, A.; Author 2, B. Book Title, 3rd ed.; Publisher: Publisher Location, Country, Year; pp. 154–196.
3. Author 1, A.; Author 2, B. Title of the chapter. In Book Title, 2nd ed.; Editor 1, A., Editor 2, B., Eds.; Publisher: Publisher Location, Country, Year; Volume 3, pp. 154–196.

ï‚·  Unpublished materials intended for publication:
4. Author 1, A.B.; Author 2, C. Title of Unpublished Work (optional). Correspondence Affiliation, City, State, Country. year, status(manuscript in preparationto be submitted).
5. Author 1, A.B.; Author 2, C. Title of Unpublished Work. Abbreviated Journal Name year, phrase indicating stage of publication(submittedacceptedin press).

ï‚·  Unpublished materials not intended for publication:
6. Author 1, A.B. (Affiliation, City, State, Country); Author 2, C. (Affiliation, City, State, Country). Phase describing the material, year. (phase: Personal communication; Private communication; Unpublished work; etc.)

ï‚·  Conference Proceedings:
7. Author 1, A.B.; Author 2, C.D.; Author 3, E.F. Title of Presentation. In Title of the Collected Work (if available), Proceedings of the Name of the Conference, Location of Conference, Country, Date of Conference; Editor 1, Editor 2, Eds. (if available); Publisher: City, Country, Year (if available); Abstract Number (optional), Pagination (optional).

ï‚·  Thesis:
8. Author 1, A.B. Title of Thesis. Level of Thesis, Degree-Granting University, Location of University, Date of Completion.

ï‚·  Websites:
9. Title of Site. Available online: URL (accessed on Day Month Year).
Unlike published works, websites may change over time or disappear, so we encourage you create an archive of the cited website using a service such as WebCite. Archived websites should be cited using the link provided as follows:
10. Title of Site. URL (archived on Day Month Year).

For example, it should be [1-3], instead of [1]-[3], in L34. Highly recommend double-checking the whole manuscript.

In the text, reference numbers should be placed in square brackets [ ], and placed before the punctuation; for example [1], [1–3] or [1,3]. For embedded citations in the text with pagination, use both parentheses and brackets to indicate the reference number and page numbers; for example [5] (p. 10). or [6] (pp. 101–105).

The ‘Author Contributions’, ‘Acknowledgements’, and ‘Conflicts of Interest’ sections still deviate from Horticulturaeformat. Since it is required as the journal guidelines, it is highly recommend double-checking the format for these three sections to better follow the horticulturae guidelines.

For research articles with several authors, a short paragraph specifying their individual contributions must be provided. The following statements should be used "Conceptualization, X.X. and Y.Y.; Methodology, X.X.; Software, X.X.; Validation, X.X., Y.Y. and Z.Z.; Formal Analysis, X.X.; Investigation, X.X.; Resources, X.X.; Data Curation, X.X.; Writing – Original Draft Preparation, X.X.; Writing – Review & Editing, X.X.; Visualization, X.X.; Supervision, X.X.; Project Administration, X.X.; Funding Acquisition, Y.Y.”, please turn to the CRediT taxonomy for the term explanation. For more background on CRediT, see here. "Authorship must include and be limited to those who have contributed substantially to the work. Please read the section concerning the criteria to qualify for authorship carefully".

Author Response

Dear.
Reviewers

Thank you for suggestions for improving our manuscript,
Hopefully our improvements are in accordance with your suggestions.

Authors
Dwi Setyorini

Reviewer 2 Report

There are still some concerns that need to be addressed to enhance the quality of the manuscript. The authors did not make any significant correction. My specific comments are as follows:

Abstract:

“The next test used is the Tukey alpha 0.05 test.” Delete

Introduction:

Add literatures on effect of ethylene on shelf life of fruit/vegetables

“Tomato mutants with mutations in the ethylene receptor gene (SlETR1) have been screened using a Micro-Tom TILLING platform.” Add citation

Based on your objectives, please compare how your study is different from those that have already been published

Materials and methods:

How many tomato samples used?

Elaborate more on the shelf life evaluation

Results and discussion:

Revise the title for all subsection in section 3 (Results)-should be in general form

Elaborate more on results for subsection 3.1

Figure 2 is not clear. Change to new image

“This is because there was no significant difference between F1 WT and F1 mutant in the genetic backgrounds (Table 3).” Elaborate on the no significant difference between mutant

The findings lack in terms of justification and major findings.

Conclusions:

Emphasize more on the main finding.

Add recommendation for future studies

General comments:

Please check the reference styles and grammar of the manuscript.

Author Response

Dear.
Reviewers

Thank you for your suggestions for improving our manuscript,
Hopefully, our improvements are in accordance with your suggestions.

Authors
Dwi Setyorini

Round 3

Reviewer 2 Report

The authors have addressed the comments. Hence, the paper can be accepted.

Author Response

Dear Reviewer

We hereby submit a revised manuscript entitled 'Effect of Ethylene Sletr1-2 Receptor Allele on flowering, fruit phenotype, yield, and shelf life of four F1 generations of Tropical tomatoes (Solanum lycopersicum L.)".

We hope that the revision of this manuscript is in accordance with the suggestions from the reviewers.

Thank you

Author,

Dwi Setyorini
